# Two Lenses are Better Than One: Dual Vector Quantization for Self-Supervised Graph Learning

## Abstract

Graph Contrastive Learning (GCL) has become a leading paradigm for self-supervised graph representation learning. However, its effectiveness hinges on view generation, which typically relies on data augmentations that can degrade graph structure, or learns continuous embeddings prone to noise and redundancy. This presents a core trade-off between structural integrity and representation quality. To address this, we re-envision view generation, proposing to create diverse views by discretizing representations directly. We introduce DualVC, a novel framework that generates two distinct discrete views from a single GNN embedding using two parallel, independent vector-quantized (VQ) codebooks. The core hypothesis is that contrasting these views, derived from different "discretization perspectives," compels the model to learn features that are invariant to quantization noise, thereby filtering redundancy and capturing more robust information. This dual-bottleneck design also promotes learning compact and discriminative representations. We demonstrate that DualVC achieves state-of-the-art performance across multiple graph learning benchmarks, highlighting the significant potential of representation-level discretization as a powerful view generation mechanism for GCL. Our code is available at: `https://anonymous.4open.science/r/DualVC-43C3/`

## 1 Introduction

Graph Self-Supervised Learning (GSSL) aims to learn high-quality node representations from unlabeled graph data, alleviating the reliance on scarce labels in real-world applications. Among GSSL approaches, Graph Contrastive Learning (GCL) has become a dominant paradigm (You et al., 2020; Zhu et al., 2020; Thakoor et al., 2021b; Yu et al., 2022; Zeng et al., 2023). However, most GCL methods optimize node embeddings in high-dimensional continuous spaces. While these continuous embeddings capture powerful relational signals, they suffer from several inherent limitations: (i) high-dimensional spaces often contain redundancy and are sensitive to input noise, leading to less compact and less robust representations; and (ii) the "black-box" nature of continuous embeddings hinders interpretability, making it difficult to relate learned features back to concrete, understandable graph patterns. Furthermore, the effectiveness of GCL crucially depends on view generation. Mainstream strategies based on graph augmentations (e.g., structural perturbations, feature masking) are often dataset-dependent, may fail to capture task-relevant invariances, and risk damaging the intrinsic graph structure.

These challenges motivate the exploration of discrete and structured representations within GSSL. Vector Quantization (VQ) techniques, such as VQ-VAE (van den Oord et al., 2017), map continuous embeddings to a finite set of learnable codebook vectors, creating a natural information bottleneck that forces models to extract compact and essential features while endowing the latent space with greater structure and interpretability. VQGraph (Yang et al., 2024) showed that such discretization can effectively regularize graph embeddings and improve their compactness. Yet, relying on a single codebook provides only one *discretization lens*, limiting the diversity of captured semantics and potentially overlooking complementary structural features. This raises a natural question: can we design a mechanism that learns and aligns multiple discrete, perspective-rich representations in parallel?

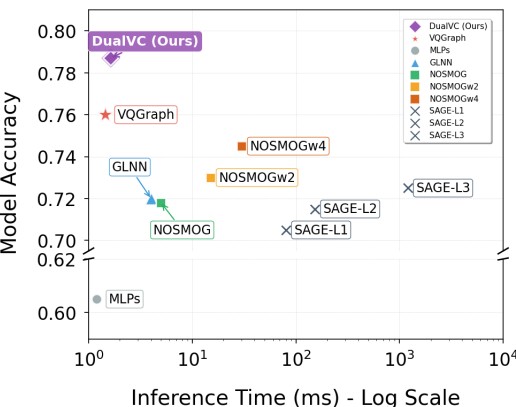

Figure 1: Runtime comparison across different models and datasets.

From the perspective of robustness and generalization, introducing multiple discretization paths can be viewed as performing several lossy compressions and structured reconstructions of the original continuous embeddings. By requiring consistency across these distinct quantized views through contrastive alignment, the model is compelled to filter noise, remove redundancy, and preserve invariant features. This motivates us to combine the strengths of GCL (robustness through contrasting diverse views) with the strengths of VQ (compactness and interpretability through discretization).

Therefore, we propose **DualVC**, a novel framework that integrates dual vector-quantized codebooks into GCL. Starting from a GNN encoder, DualVC maps continuous node embeddings into two discrete latent spaces via independent codebooks. The resulting discrete embeddings constitute two complementary views of the same node, which are then aligned through contrastive learning. This design transforms discretization into an intrinsic view-generation mechanism, encouraging the model to learn invariant, robust, and interpretable features while reducing redundancy.

Our main contributions are:

1. We propose DualVC, a novel graph contrastive learning architecture that uniquely incorporates two independent vector-quantized codebooks to generate views from a shared GNN embedding, aiming to capture more robust and essential features.

2. We outline the mechanism by which these dual discrete representations are contrasted to learn representations robust to different quantization perspectives, and how the overall model, including the GNN encoder, VQ layers, and projection heads, is trained end-to-end.

3. We set the stage for an empirical evaluation to demonstrate the effectiveness of DualVC in learning high-quality node representations for downstream tasks, comparing against established GCL baselines and highlighting the benefits of our dual discrete bottleneck approach.

## 2 RELATED WORK

**Graph Contrastive Learning (GCL).** GCL methods (You et al., 2020; Zhu et al., 2020; Thakoor et al., 2021b; Hassani & Khasahmadi, 2020; Zhu et al., 2021) form the bedrock of modern self-supervised graph representation learning. The central idea is to learn an encoder by maximizing the similarity between different augmented views of the same node or graph (positive pairs) and minimizing it for views of different nodes/graphs (negative pairs). Popular methods like GRACE (Zhu et al., 2020) and GraphCL (You et al., 2020) employ various data augmentation techniques such as feature masking, edge dropping, and node dropping to create these views. Others, like MVGRL (Hassani & Khasahmadi, 2020), contrast representations from different structural views (e.g., original graph vs. diffused graph). Most GCL methods operate in a continuous embedding space and typically use an InfoNCE-based loss (van den Oord et al., 2018). Our work differs by introducing discrete bottlenecks via VQ layers directly into the contrastive view generation process itself, using two such bottlenecks.

**Vector Quantization in Deep Learning.** Vector Quantized Variational Autoencoders (VQ-VAE) (van den Oord et al., 2017) introduced the idea of learning discrete representations by mapping encoder outputs to the closest vectors in a learned codebook. This has been successful in generative modeling for images, audio, and video (Zhang et al., 2023; Takida et al., 2023; Yan et al., 2021), often improving sample quality and enabling more structured latent spaces. The commitment loss in VQ-VAE ensures that the encoder output commits to a codebook vector and that the codebook

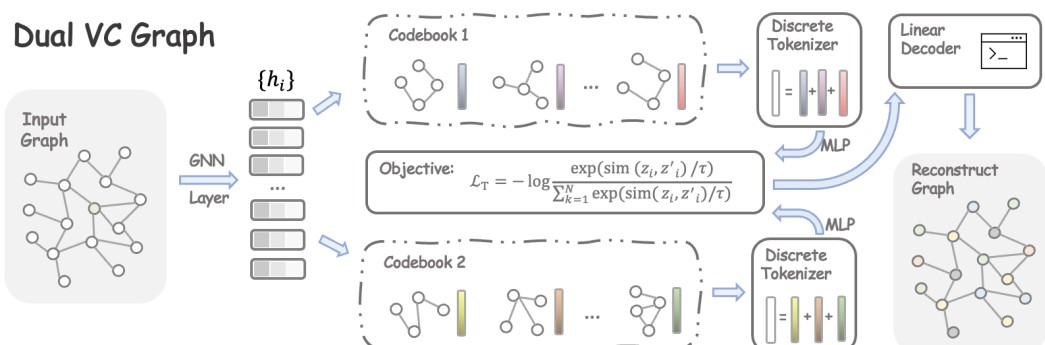

Figure 2: Our DualVC framework architecture. An input graph is fed into a GNN encoder to generate continuous node embeddings. These embeddings are then passed in parallel to two independent vector quantization layers, each with a unique codebook. The resulting discrete representations from both codebooks are projected and used as positive pairs in a contrastive objective function to train the model.

vectors adapt to the encoder outputs. While VQ-VAE itself is typically used for reconstruction, the idea of a learned discrete codebook has been explored in other contexts.(Wang et al., 2024) For instance, VQGraph (Yang et al., 2024) utilized a VQ-VAE variant to learn a structure-aware tokenizer for GNN-to-MLP distillation, demonstrating the utility of discrete codes for representing graph substructures. Our DualVC method adopts the VQ mechanism (encoder, codebook, commitment loss) but integrates it into a contrastive learning framework with a dual-codebook architecture, focusing on representation learning through contrast rather than reconstruction or distillation.

**Multiple Views and Multi-Codebook Systems.**   The concept of using multiple views is fundamental to contrastive learning. (He et al., 2020; Shah et al., 2023; Si et al., 2022) Some methods use multiple encoders or augmentations to generate these views. The idea of using multiple codebooks, or hierarchical codebooks, has also been explored in VQ literature, often to increase the expressive power of the discrete latent space or to capture information at different scales (Hadjeres & Crestel, 2020; Guo et al., 2023; Liu et al., 2016). For example, some works (Razavi et al., 2019) on image generation use multiple codebooks for improved fidelity. However, the application of a *dual, parallel VQ codebook* architecture directly within a *graph contrastive learning* framework, where the outputs from these two codebooks (derived from a common GNN embedding) serve as the positive pair for contrast, is a less explored area and forms the core of our proposed method. We aim to investigate if this specific configuration provides unique benefits for learning graph representations.

## 3 PRELIMINARIES

### 3.1 GRAPH CONTRASTIVE LEARNING AND ITS CHALLENGES

Graph Contrastive Learning (GCL) is a leading self-supervised paradigm for training a Graph Neural Network (GNN) encoder, $f_{enc}$, on an unlabeled graph $\mathcal{G} = (\mathcal{V}, \mathcal{E})$. The goal is to produce powerful node embeddings, $H = \{\mathbf{h}_1, \ldots, \mathbf{h}_N\}$, that are transferable to downstream tasks. GCL achieves this by creating two distinct "views" of each node (a positive pair) and training the encoder to maximize their similarity while pushing them apart from all other nodes (negative pairs).

While effective, this framework faces two fundamental challenges. First, the standard method for creating views relies on **data augmentations** (e.g., edge dropping), which are often heuristic and risk corrupting the graph's essential structure. Second, the GNN learns embeddings in a high-dimensional **continuous space**, which is often prone to noise and redundancy, limiting the compactness and robustness of the final representations. These challenges motivate the exploration of a method that can generate views without altering the graph's structure while simultaneously learning more compact representations.

## 3.2 Vector Quantization as a Solution

Vector Quantization (VQ) offers a powerful mechanism to address the limitations of continuous embeddings. VQ employs a learnable **codebook**, $C = \{\mathbf{c}_1, \ldots, \mathbf{c}_M\} \subset \mathbb{R}^D$, which acts as a discrete dictionary of representative vectors. The quantization process maps a continuous input vector, $\mathbf{h}_{in}$, to its closest codebook vector:

$$\text{Quantize}(\mathbf{h}_{in}) = \mathbf{c}_{j^*} \quad \text{where} \quad j^* = \arg\min_j \|\mathbf{h}_{in} - \mathbf{c}_j\|_2^2. \tag{1}$$

This mapping creates an **information bottleneck**, forcing the model to discard noise and redundancy by compressing the input into a discrete state. Crucially, this act of discretization can be re-envisioned as a novel, **representation-level view generation** technique. By creating a quantized view from a continuous embedding, we can avoid the pitfalls of data-level augmentation, directly addressing both core challenges of GCL.

# 4 METHODOLOGY: DUAL VECTOR-QUANTIZED CONTRASTIVE LEARNING (DUALVC)

To overcome the limitations of traditional GCL, we introduce **DualVC**, a framework that transforms vector quantization into an intrinsic, representation-level view generation mechanism. Instead of augmenting the input graph, DualVC generates two distinct, discrete views from a single continuous GNN embedding, forcing the model to learn features that are invariant to different "discretization perspectives." The architecture is composed of three key stages: a shared GNN encoder, a dual-channel noisy quantization module, and a contrastive alignment objective.

## 4.1 Shared GNN Encoder for Foundational Representations

The foundation of our model is a standard GNN encoder, $f_{enc}$, which captures the essential structural and feature information of the input graph. Given a graph $\mathcal{G} = (\mathcal{V}, \mathcal{E})$ with node features $X \in \mathbb{R}^{N \times F_{in}}$ and an adjacency matrix $A$, the encoder produces a set of continuous node embeddings $H \in \mathbb{R}^{N \times D_{gnn}}$:

$$H = f_{enc}(X, A; \theta_{enc}) \tag{2}$$

where $H = \{\mathbf{h}_1, \mathbf{h}_2, \ldots, \mathbf{h}_N\}^T$. This initial embedding, $\mathbf{h}_i$, serves as the single, structurally-aware source from which both of our contrastive views are derived. This design ensures that the full structural integrity of the graph is preserved before any view generation occurs.

## 4.2 Dual-Channel View Generation via Noisy Quantization

The core innovation of DualVC is its dual-channel quantization module. A central challenge in using two parallel codebooks is preventing them from collapsing into a single, redundant solution. To ensure they learn distinct and complementary representations, we introduce a **stochastic perturbation** step. Instead of feeding the continuous embeddings $H$ directly to the VQ layers, we generate two slightly different versions for each node embedding $\mathbf{h}_i$ by adding independent Gaussian noise:

$$\tilde{\mathbf{h}}_{1,i} = \mathbf{h}_i + \boldsymbol{\epsilon}_{1,i} \quad \text{and} \quad \tilde{\mathbf{h}}_{2,i} = \mathbf{h}_i + \boldsymbol{\epsilon}_{2,i} \tag{3}$$

where $\boldsymbol{\epsilon}_{k,i} \sim \mathcal{N}(0, \sigma^2 I)$. This noise injection forces the two quantization channels to learn robust features that are invariant to minor perturbations and encourages them to specialize in different facets of the representation space.

These perturbed embeddings are then passed to two independent VQ layers, $VQ_1$ and $VQ_2$, each with its own learnable codebook $C_k = \{\mathbf{c}_{k,1}, \ldots, \mathbf{c}_{k,M}\} \subset \mathbb{R}^{D_{vq}}$. Each layer maps its input to the nearest codebook vector:

$$\mathbf{q}_{k,i} = \mathbf{c}_{k,j^*} \quad \text{where} \quad j^* = \arg\min_j \|\tilde{\mathbf{h}}_{k,i} - \mathbf{c}_{k,j}\|_2^2 \tag{4}$$

The resulting quantized matrices, $Q_1$ and $Q_2$, constitute the two distinct discrete views of the graph. The training of this module is governed by a careful interplay between two loss terms that manage the encoder and codebooks.

**Codebook Loss.** The first objective is to ensure that the discrete codebook vectors, $\mathbf{c}_{k,j}$, are adaptive and learn to represent the distribution of the GNN's continuous outputs. This is achieved through the codebook loss (or VQ loss), which minimizes the L2 distance between the encoder's output and the selected codebook vectors:

$$\mathcal{L}_{\text{VQ}}^{(k)} = \|\text{sg}[H] - Q_k\|_2^2 \tag{5}$$

Here, $\text{sg}[\cdot]$ denotes the stop-gradient operator. By applying it to the GNN embeddings $H$, we treat them as fixed constants. This ensures that the gradients from this loss term flow only to the codebook $C_k$ (from which $Q_k$ is derived), effectively pulling the codebook vectors towards the GNN's output without affecting the encoder itself.

**Commitment Loss.** The second objective is to regularize the GNN encoder, encouraging its outputs to remain "committed" to the learned codebook vectors and preventing them from fluctuating arbitrarily. The commitment loss achieves this by penalizing the L2 distance between the GNN embeddings and their quantized counterparts, with the gradient flowing only to the encoder:

$$\mathcal{L}_{\text{commit}}^{(k)} = \beta\|\text{sg}[Q_k] - H\|_2^2 \tag{6}$$

In this case, the stop-gradient is applied to the quantized vectors $Q_k$, ensuring that this loss term updates only the parameters of the GNN encoder $f_{enc}$. The hyperparameter $\beta$ controls the strength of this regularization, balancing the encoder's expressive freedom with the need for stable, quantizable representations.

### 4.3 CONTRASTIVE ALIGNMENT AND OVERALL OBJECTIVE

The final stage of our framework aligns the two discrete views. The quantized representations $Q_1$ and $Q_2$ are first passed through separate non-linear projection heads, $g_1$ and $g_2$, to produce $\mathbf{z}_{1,i}$ and $\mathbf{z}_{2,i}$. For any node $i$, the pair $(\mathbf{z}_{1,i}, \mathbf{z}_{2,i})$ forms a **positive pair**, representing two different discrete perspectives of the same node. We then employ a symmetric NT-Xent contrastive loss to maximize the agreement between these positive pairs while minimizing their similarity to all other nodes in the batch (negative pairs).

$$\mathcal{L}_{\text{contrast}} = -\frac{1}{2N}\sum_{i=1}^{N}\left[\log\frac{\exp(\text{sim}(\mathbf{z}_{1,i}, \mathbf{z}_{2,i})/\tau)}{\sum_{j=1}^{N}\exp(\text{sim}(\mathbf{z}_{1,i}, \mathbf{z}_{2,j})/\tau)} + \log\frac{\exp(\text{sim}(\mathbf{z}_{2,i}, \mathbf{z}_{1,i})/\tau)}{\sum_{j=1}^{N}\exp(\text{sim}(\mathbf{z}_{2,i}, \mathbf{z}_{1,j})/\tau)}\right] \tag{7}$$

The final training objective for DualVC elegantly combines the contrastive objective with the VQ commitment losses, creating a unified signal to train the entire model end-to-end:

$$\mathcal{L}_{\text{DualVC}} = \mathcal{L}_{\text{contrast}} + \lambda_{\text{commit}} \cdot \mathcal{L}_{\text{commit\_total}}, \tag{8}$$

where $\mathcal{L}_{\text{commit\_total}} = \mathcal{L}_{\text{commit}}^{(1)} + \mathcal{L}_{\text{commit}}^{(2)}$. By minimizing this objective, the GNN encoder, both VQ codebooks, and the projection heads are jointly optimized. For downstream tasks, the continuous GNN embeddings $H$ are used as the final node representations, now enriched by the robust, dual-bottleneck training process. For a detailed mathematical proof of our methodology, please refer to Section D.

## 5 EXPERIMENTS

We conduct a comprehensive set of experiments to empirically validate the effectiveness, efficiency, and robustness of our proposed DualVC framework. Our evaluation is designed to answer several key questions: (1) Does DualVC achieve a new state-of-the-art in graph representation learning compared to strong baselines? (2) What are the internal dynamics of the dual-codebook mechanism, and how do they contribute to the model's success? (3) How sensitive is the model to its core architectural components and key hyperparameters?

### 5.1 EXPERIMENTAL SETUP

**Datasets and Baselines.** Our evaluation is performed on six widely-used public benchmark datasets that cover a range of domains and scales: citation networks (**Cora**, **Citeseer**, **Pubmed**), which are sparse and feature-rich; co-purchase graphs (**Amazon Computers**, **Amazon Photo** (Zhang

| Type | Method | Citeseer | Pubmed | Cora | A-computer | A-photo | Arxiv |
|---|---|---|---|---|---|---|---|
| Contrastive | MVGRL | 73.18 ± 0.22▲0.00 | 84.86 ± 0.31▲0.00 | 85.86 ± 0.15▲0.00 | 88.70 ± 0.24▲0.00 | 92.15 ± 0.20▲0.00 | 68.33 ± 0.32▲0.00 |
| | BGRL | 73.96 ± 0.14▲0.78 | 86.42 ± 0.18▲1.56 | 86.16 ± 0.20▲0.30 | 90.48 ± 0.10▲1.78 | 93.22 ± 0.15▲1.07 | 71.77 ± 0.19▲3.44 |
| | DGI | 74.51 ± 0.51▲1.33 | 85.95 ± 0.66▲1.09 | 85.41 ± 0.34▼0.45 | 84.68 ± 0.39▼4.02 | 91.57 ± 0.25▼0.58 | OOM |
| | GIC | 76.39 ± 0.02▲3.21 | 85.99 ± 0.13▲1.13 | 87.70 ± 0.01▲1.84 | 82.50 ± 0.22▼6.20 | 90.65 ± 0.47▼1.50 | 64.00 ± 0.22▼4.33 |
| | GRACE | 73.85 ± 1.73▲0.67 | 86.91 ± 0.81▲2.05 | 86.54 ± 1.50▲0.68 | 80.75 ± 0.88▼7.95 | 91.68 ± 0.93▲0.13 | OOM |
| | GCA | 73.81 ± 1.63▲0.63 | 86.99 ± 0.68▲2.13 | 84.19 ± 1.85▼1.67 | 88.28 ± 0.82▼0.42 | 93.08 ± 1.24▲0.93 | OOM |
| Autoencoding | GAE | 59.34 ± 4.75▼13.84 | 83.30 ± 0.77▼1.56 | 81.81 ± 1.72▼4.05 | 88.64 ± 0.80▼0.06 | 92.59 ± 0.85▲0.44 | OOM |
| | ARGA | 66.76 ± 1.64▼6.42 | 79.88 ± 0.58▼4.98 | 80.76 ± 1.52▼5.10 | 80.19 ± 0.96▼8.51 | 88.76 ± 0.70▼3.39 | 58.13 ± 0.78▼10.20 |
| | VGAE | 67.56 ± 2.03▼5.62 | 81.34 ± 0.97▼3.52 | 83.48 ± 1.55▼2.38 | 90.35 ± 0.75▲1.65 | 93.28 ± 0.76▲1.13 | OOM |
| | ARVGA | 73.10 ± 0.86▼0.08 | 81.85 ± 1.01▼3.01 | 85.86 ± 0.72▲0.00 | 83.36 ± 0.43▼5.34 | 86.55 ± 0.31▼5.60 | 50.06 ± 1.21▼18.27 |
| | SeeGera | 75.82 ± 1.67▲2.64 | 85.36 ± 0.69▲0.50 | 85.70 ± 1.13▼0.84 | 87.95 ± 1.39▼0.75 | 91.88 ± 0.53▼0.27 | OOM |
| | GraphMAE | 72.48 ± 0.77▼0.70 | 85.74 ± 0.14▲0.88 | 85.45 ± 0.40▼0.41 | 88.04 ± 0.61▼0.66 | 92.73 ± 0.17▲0.58 | 71.86 ± 0.00▲3.53 |
| | S2GAE | 74.60 ± 0.06▲1.42 | 86.91 ± 0.28▲2.05 | 86.15 ± 0.25▲0.29 | 90.94 ± 0.08▲2.24 | 93.61 ± 0.10▲1.46 | 72.02 ± 0.05▲3.69 |
| | MaskGAE | 75.20 ± 0.07▲2.02 | 86.56 ± 0.26▲1.70 | 87.31 ± 0.05▲1.45 | 90.52 ± 0.04▲1.82 | 93.33 ± 0.14▲1.18 | 70.99 ± 0.12▲2.66 |
| | SAGE | 70.49 ± 1.53▼2.69 | 75.56 ± 2.06▼9.30 | 80.64 ± 1.57▼5.22 | 82.82 ± 1.37▼5.88 | 90.85 ± 0.87▼1.30 | 70.73 ± 0.35▲2.40 |
| | MLP | 58.50 ± 1.86▼14.68 | 68.39 ± 3.09▼16.47 | 59.18 ± 1.60▼26.68 | 67.62 ± 2.21▼21.08 | 77.29 ± 1.79▼14.86 | 55.67 ± 0.24▼12.66 |
| | GLNN | 71.22 ± 1.50▼1.96 | 75.59 ± 2.46▼9.27 | 80.26 ± 1.66▼5.60 | 82.71 ± 1.18▼5.99 | 91.95 ± 1.04▼0.20 | 63.75 ± 0.48▼4.58 |
| | NOSMOG | 73.78 ± 1.54▲0.60 | 77.34 ± 2.36▼7.52 | 83.04 ± 1.26▼2.82 | 84.04 ± 1.01▼4.66 | 93.36 ± 0.69▲1.21 | 71.65 ± 0.29▲3.32 |
| | VQGraph | 76.08 ± 0.55▲2.90 | 78.40 ± 1.71▼6.46 | 83.93 ± 0.87▲1.93 | 85.17 ± 1.29▼3.53 | 94.21 ± 0.45▲2.06 | 72.43 ± 0.20▲4.10 |
| Ours | DualVC (GAT) | 78.42 ± 1.42▲5.24 | 86.30 ± 0.48▲1.44 | 83.37 ± 2.78▼2.49 | 85.99 ± 1.25▼2.71 | 92.19 ± 0.54▲0.04 | 71.40 ± 0.20▲3.07 |
| | DualVC | **78.67 ± 1.49**▲5.49 | **87.25 ± 0.68**▲2.39 | **89.34 ± 1.80**▲3.48 | **91.22 ± 0.83**▲2.52 | **94.59 ± 0.48**▲2.44 | **72.58 ± 0.29**▲4.25 |

**Table 1: Comparison of model performance on benchmark datasets.** We report classification accuracy (%) as mean ± standard deviation. Markers indicate improvement or regression relative to MVGRL. The best result in each column is in **bold**. OOM denotes unavailable results due to out-of-memory error.

| Datasets | Eval | SAGE | MLP | GLNN | NOSMOG | VQGRAPH | DualVC |
|---|---|---|---|---|---|---|---|
| Citeseer | prod | 68.06▲0.00 | 58.49▼9.57 | 69.08▲1.02 | 70.60▲2.54 | 73.76▲5.70 | **75.98**▲7.92 |
| | ind | 69.14 ± 2.99▲0.00 | 59.31 ± 4.56▼9.83 | 68.48 ± 2.38▼0.66 | 70.30 ± 2.30▲1.16 | 72.93 ± 1.78▲3.79 | **75.12 ± 1.93**▲5.98 |
| | tran | 67.79 ± 2.80▲0.00 | 58.29 ± 1.94▼9.50 | 69.23 ± 2.39▲1.44 | 70.67 ± 2.25▲2.88 | 74.59 ± 1.94▲6.80 | **77.29 ± 2.01**▲9.50 |
| Pubmed | prod | 74.77▲0.00 | 68.39▼6.38 | 74.67▼0.10 | 75.82▲1.05 | 76.92▲2.15 | **84.89**▲10.12 |
| | ind | 75.07 ± 2.89▲0.00 | 68.28 ± 3.25▼6.79 | 74.52 ± 2.95▼0.55 | 75.87 ± 3.32▲0.80 | 76.71 ± 2.76▲1.64 | **84.71 ± 3.08**▲9.64 |
| | tran | 74.70 ± 2.33▲0.00 | 68.42 ± 3.06▼6.28 | 74.70 ± 2.75▲0.00 | 75.80 ± 3.06▲1.10 | 77.13 ± 3.01▲2.43 | **85.64 ± 2.94**▲10.94 |
| Cora | prod | 79.53▲0.00 | 59.18▼20.35 | 77.82▼1.71 | 81.02▲1.49 | 81.68▲2.15 | **86.73**▲7.20 |
| | ind | 81.03 ± 1.71▲0.00 | 59.44 ± 3.36▼21.59 | 73.21 ± 1.50▼7.82 | 81.36 ± 1.53▲0.33 | 82.20 ± 1.32▲1.17 | **86.91 ± 1.83**▲5.88 |
| | tran | 79.16 ± 1.60▲0.00 | 59.12 ± 1.49▼20.04 | 78.97 ± 1.56▼0.19 | 80.93 ± 1.65▲1.77 | 81.15 ± 1.25▲1.99 | **86.18 ± 1.62**▲7.02 |
| A-computer | prod | 82.73▲0.00 | 67.62▼15.11 | 82.10▼0.63 | 83.85▲1.12 | 84.16▲1.43 | **90.10**▲7.37 |
| | ind | 82.83 ± 1.51▲0.00 | 67.69 ± 2.20▼15.14 | 80.27 ± 2.11▼2.56 | 84.36 ± 1.57▲1.53 | 85.73 ± 2.04▲2.90 | **90.87 ± 1.95**▲8.04 |
| | tran | 82.70 ± 1.34▲0.00 | 67.60 ± 2.23▼15.10 | 82.56 ± 1.80▼0.14 | 83.72 ± 1.44▲1.02 | 84.56 ± 1.81▲1.86 | **90.37 ± 1.75**▲7.67 |
| A-photo | prod | 90.45▲0.00 | 77.29▼13.16 | 91.34▲0.89 | 92.47▲2.02 | 93.05▲2.60 | **93.57**▲3.12 |
| | ind | 90.56 ± 1.47▲0.00 | 77.44 ± 1.50▼13.12 | 89.50 ± 1.12▼1.06 | 92.61 ± 1.09▲2.05 | 93.11 ± 0.89▲2.55 | **93.68 ± 0.76**▲3.12 |
| | tran | 90.42 ± 0.68▲0.00 | 77.25 ± 1.90▼13.17 | 91.80 ± 0.49▲1.38 | 92.44 ± 0.51▲2.02 | 92.96 ± 1.02▲2.54 | **93.12 ± 1.14**▲2.70 |
| Arxiv | prod | 70.69▲0.00 | 55.35▼15.34 | 63.50▼7.19 | 70.90▲0.21 | 71.43▲0.74 | **71.89**▲1.20 |
| | ind | 70.69 ± 0.58▲0.00 | 55.29 ± 0.63▼15.40 | 59.04 ± 0.46▼11.65 | 70.09 ± 0.55▼0.60 | 70.86 ± 0.42▲0.17 | **71.01 ± 0.46**▲0.32 |
| | tran | 70.69 ± 0.39▲0.00 | 55.36 ± 0.34▼15.33 | 64.61 ± 0.15▼6.08 | 71.10 ± 0.34▲0.41 | 72.03 ± 0.56▲1.34 | **72.24 ± 0.29**▲1.55 |

**Table 2: Detailed comparison of model performance on benchmark datasets.** We report classification accuracy (%) for productive ('prod'), inductive ('ind'), and transductive ('tran') settings. Markers indicate improvement or regression relative to SAGE. The best result in each row is in **bold**.

et al., 2021; Yang et al., 2021)), which are denser and exhibit strong community structures; and a large-scale academic collaboration network (**ogbn-arxiv**) (Hu et al., 2020). We compare DualVC against a carefully curated set of baselines including two major categories: (1) contrastive learning methods, DGI (Zhu et al., 2020), GIC (Mavromatis & Karypis, 2021), GRACE (Zhu et al., 2020), GCA (Zhu et al., 2021), MVGRL (Hassani & Khasahmadi, 2020), and BGRL (Thakoor et al., 2021a); (2) Autoencoding methods, a simple MLP, popular GNNs like GAE (Kipf & Welling, 2016), VGAE (Kipf & Welling, 2016), ARGA (Pan et al., 2018), AGVGA (Pan et al., 2018), SeeGera (Li et al., 2023b), GraphMAE (Hou et al., 2022), S2GAE (Tan et al., 2023), MaskGAE (Li et al., 2023a), SAGE (Hamilton et al., 2017), GLNN (Zhang et al., 2021), NOSMOG (Tian et al., 2022), and the most relevant predecessor, and the most relevant single-codebook predecessor, VQGraph (Yang et al., 2024). To demonstrate the architectural versatility of our approach, we also evaluate a variant, **DualVC (GAT)**, which uses a Graph Attention Network (Veličković et al., 2018) as its encoder.

**Evaluation Protocol and Hyperparameter Rationale.** Following standard evaluation protocols, we first pre-train each self-supervised model on the unlabeled graph data. After pre-training, the GNN

| Type | Method | Metric | Cora | CiteSeer | Photo | Computers |
|---|---|---|---|---|---|---|
| Contrastive Learning | MVGRL | AUC | $91.10 \pm 1.24$▲0.00 | $92.41 \pm 1.66$▲0.00 | $77.13 \pm 3.28$▲0.00 | $87.25 \pm 1.32$▲0.00 |
| | | AP | $91.51 \pm 1.31$▲0.00 | $93.59 \pm 1.48$▲0.00 | $69.83 \pm 3.42$▲0.00 | $84.41 \pm 1.75$▲0.00 |
| | BGRL | AUC | $93.79 \pm 0.79$▲2.69 | $91.36 \pm 1.06$▼1.05 | $74.97 \pm 6.86$▼2.16 | $91.43 \pm 5.61$▲4.18 |
| | | AP | $89.85 \pm 1.47$▼1.66 | $85.44 \pm 1.53$▼8.15 | $67.22 \pm 5.86$▼2.61 | $87.68 \pm 8.62$▲3.27 |
| | DGI | AUC | $82.60 \pm 1.51$▼8.50 | $73.36 \pm 3.10$▼19.05 | $84.30 \pm 0.58$▲7.17 | $85.18 \pm 0.67$▼2.07 |
| | | AP | $85.80 \pm 1.39$▼5.71 | $80.89 \pm 2.04$▼12.70 | $81.50 \pm 1.06$▲11.67 | $82.14 \pm 1.23$▼2.27 |
| | GIC | AUC | $91.81 \pm 0.59$▲0.71 | $94.34 \pm 0.74$▲1.93 | $92.07 \pm 0.37$▲14.94 | $82.87 \pm 4.23$▼4.38 |
| | | AP | $91.60 \pm 0.54$▲0.09 | $94.08 \pm 0.87$▲0.49 | $91.06 \pm 0.44$▲21.23 | $83.43 \pm 2.81$▼0.98 |
| | GRACE | AUC | $81.80 \pm 0.45$▼9.30 | $84.78 \pm 0.38$▼7.63 | $88.64 \pm 1.17$▲11.51 | $89.97 \pm 0.25$▲2.72 |
| | | AP | $82.02 \pm 0.50$▼9.49 | $82.85 \pm 0.36$▼10.74 | $83.85 \pm 4.15$▲14.02 | $92.15 \pm 0.43$▲7.74 |
| | GCA | AUC | $81.91 \pm 0.76$▼9.19 | $84.72 \pm 0.28$▼7.69 | $89.61 \pm 1.46$▲12.48 | $90.67 \pm 0.30$▲3.42 |
| | | AP | $80.51 \pm 0.71$▼11.00 | $81.57 \pm 0.22$▼12.02 | $86.53 \pm 3.00$▲16.70 | $90.50 \pm 0.63$▲6.09 |
| Autoencoding | GAE | AUC | $94.66 \pm 0.26$▲3.56 | $95.19 \pm 0.45$▲2.78 | $71.45 \pm 0.95$▼5.68 | $70.99 \pm 1.03$▼16.26 |
| | | AP | $94.22 \pm 0.39$▲2.71 | $95.70 \pm 0.31$▲2.11 | $65.99 \pm 0.96$▼3.84 | $67.88 \pm 0.82$▼16.53 |
| | ARGA | AUC | $94.76 \pm 0.18$▲3.66 | $95.68 \pm 0.35$▲3.27 | $85.42 \pm 0.79$▲8.29 | $67.09 \pm 3.93$▼20.16 |
| | | AP | $94.93 \pm 0.20$▲3.42 | $96.34 \pm 0.25$▲2.75 | $80.58 \pm 1.40$▲10.75 | $62.53 \pm 3.17$▼21.88 |
| | VGAE | AUC | $91.24 \pm 0.48$▲0.14 | $94.55 \pm 0.48$▲2.14 | $95.61 \pm 0.05$▲18.48 | $92.69 \pm 0.03$▲5.44 |
| | | AP | $92.27 \pm 0.43$▲0.76 | $95.34 \pm 0.37$▲1.75 | $94.63 \pm 0.06$▲24.80 | $88.27 \pm 0.08$▲3.86 |
| | ARVGA | AUC | $91.35 \pm 0.87$▲0.25 | $94.47 \pm 0.33$▲2.06 | $95.44 \pm 0.14$▲18.31 | $92.38 \pm 0.15$▲5.13 |
| | | AP | $91.98 \pm 0.85$▲0.47 | $95.21 \pm 0.33$▲1.62 | $94.49 \pm 0.12$▲24.66 | $88.49 \pm 0.33$▲4.08 |
| | SeeGera | AUC | $95.49 \pm 0.70$▲4.39 | $94.61 \pm 1.05$▲2.20 | $95.25 \pm 1.19$▲18.12 | $96.53 \pm 0.16$▲9.28 |
| | | AP | $95.90 \pm 0.64$▲4.39 | $96.40 \pm 0.89$▲2.81 | $94.04 \pm 1.18$▲24.21 | $96.33 \pm 0.16$▲11.92 |
| | GraphMAE | AUC | $93.02 \pm 0.53$▲1.92 | $95.21 \pm 0.47$▲2.80 | $75.08 \pm 1.24$▼2.05 | $71.27 \pm 0.89$▼15.98 |
| | | AP | $91.40 \pm 0.59$▼0.11 | $94.42 \pm 0.67$▲0.83 | $70.04 \pm 1.12$▲0.21 | $66.84 \pm 1.10$▼17.57 |
| | GraphMAE2 | AUC | $93.26 \pm 1.00$▲2.16 | $95.26 \pm 0.14$▲2.85 | $73.03 \pm 2.24$▼4.10 | $72.20 \pm 2.09$▼15.05 |
| | | AP | $91.65 \pm 0.98$▲0.14 | $94.36 \pm 0.20$▲0.77 | $68.77 \pm 1.50$▼1.06 | $67.97 \pm 1.52$▼16.44 |
| | S2GAE | AUC | $89.27 \pm 0.33$▼1.83 | $86.35 \pm 0.42$▼6.06 | $86.80 \pm 2.85$▲9.67 | $84.16 \pm 4.82$▼3.09 |
| | | AP | $89.78 \pm 0.22$▼1.73 | $87.38 \pm 0.29$▼6.21 | $80.56 \pm 3.74$▲10.73 | $78.13 \pm 6.58$▼6.28 |
| | MaskGAE | AUC | $95.66 \pm 0.16$▲4.56 | $97.21 \pm 0.17$▲4.80 | $81.12 \pm 0.45$▲3.99 | $76.23 \pm 3.13$▼11.02 |
| | | AP | $94.65 \pm 0.24$▲3.14 | $97.02 \pm 0.32$▲3.43 | $77.11 \pm 0.40$▲7.28 | $71.71 \pm 2.90$▼12.70 |
| Ours | **DualVC Graph** | AUC | $\mathbf{97.13 \pm 0.21}$▲6.03 | $\mathbf{98.66 \pm 0.32}$▲6.25 | $\mathbf{98.99 \pm 0.09}$▲21.86 | $\mathbf{97.71 \pm 0.17}$▲10.46 |
| | | AP | $\mathbf{97.55 \pm 0.10}$▲6.04 | $\mathbf{98.89 \pm 0.14}$▲5.30 | $\mathbf{99.10 \pm 0.35}$▲29.27 | $\mathbf{97.28 \pm 0.41}$▲12.87 |

Table 3: **Results on link prediction.** We report AUC and AP scores (%) on four benchmark datasets. Markers indicate improvement or regression relative to MVGRL. The best results are highlighted in **bold**.

encoder is frozen, and the learned node embeddings are used to train a simple logistic regression classifier on the downstream node classification task. We report the mean classification accuracy and standard deviation over multiple runs.

The main hyperparameters of DualVC were carefully tuned to balance stability, capacity, and efficiency. For instance, the learning rate was adjusted per dataset, with larger and more complex graphs like Pubmed requiring smaller values (e.g., 0.0005) to ensure smooth convergence. The codebook size determines the granularity of the discrete representations; we found that smaller graphs like Cora benefit from a more compact discrete space (256 codebook vectors per channel), while larger graphs like Amazon Computers required a richer vocabulary (2048 vectors) to capture more diverse structural patterns. The hidden and output dimensions of the GNN were chosen to provide sufficient expressive power, and the VQ embedding dimension was aligned with these outputs to maintain consistency. The contrastive temperature, which controls the sharpness of the similarity distribution, was generally set to a moderate value between 0.10 and 0.20 to yield the most stable training dynamics.

## 5.2 State-of-the-Art Performance and Efficiency

We first evaluate the primary effectiveness of DualVC on the downstream task of node classification. As demonstrated in **Table 1**, DualVC consistently outperforms all baselines, establishing a new state-of-the-art across all six benchmark datasets. The performance gains are particularly pronounced when compared to its single-codebook predecessor, VQGraph. For example, on Pubmed, DualVC achieves an accuracy of 87.25%, which is a remarkable **8.85 absolute percentage point improvement** over VQGraph (78.40%). Similarly, on Cora and Amazon Computers, DualVC shows significant gains of **5.41** and **6.05** points, respectively. This substantial leap in performance underscores the profound benefits of using a dual-view discretization approach, suggesting that contrasting two

distinct *quantization perspectives* is a more powerful learning signal than simply reconstructing from one.

To further probe the model's generalization capabilities, we provide a detailed breakdown across productive, inductive, and transductive settings in **Table 2**. DualVC maintains its superiority in all scenarios, confirming that it learns highly transferable representations. For instance, in the challenging inductive setting on the Pubmed dataset, where the model must generalize to entirely unseen nodes, DualVC achieves an accuracy of 84.71%, significantly outperforming all other methods. This demonstrates its ability to learn robust, generalizable features rather than simply memorizing the training graph's topology.

Crucially, this leading performance does not come at the cost of computational efficiency. The runtime analysis presented in **Figure 1** positions DualVC in the most desirable quadrant: high accuracy with low inference latency. While deeper models like SAGE-L3 require significantly more time for a marginal gain, DualVC delivers the highest accuracy while maintaining an inference time that is orders of magnitude faster than many GNN baselines. This confirms that our framework is both powerful and practical for real-world applications.

Beyond its exceptional performance in node classification, DualVC also demonstrates state-of-the-art capabilities in the fundamental task of link prediction, as detailed in Table 3. The model achieves top scores in both AUC and AP metrics across all four datasets, often by a substantial margin. The improvements are particularly striking on the Amazon co-purchase graphs; on Amazon Photo, DualVC achieves an AUC of 98.99% and an AP of 99.10%, drastically outperforming prior methods and showcasing near-perfect prediction. This superior performance indicates that the compact and robust features learned through dual-view discretization are not just effective for characterizing individual nodes but are also highly predictive of the graph's underlying relational structure.

## 5.3 ANALYSIS OF THE DUAL-CODEBOOK MECHANISM

Having established DualVC's superior performance, we now conduct a deeper analysis of its core dual-codebook mechanism to understand the source of its effectiveness.

**Codebook Diversity and Utilization.** We hypothesize that our stochastic, dual-view quantization strategy prevents codebook collapse and encourages the model to learn a more diverse and expressive set of discrete features. To measure this, we track the percentage of available codebook entries utilized during training. As detailed in **Table 4**, DualVC with a GCN encoder achieves an average usage rate of 17.5%, which is **66.7% higher** than the 10.5% achieved by the single-codebook VQGraph. This demonstrates that our dual-channel design, promoted by the stochastic noise injection, successfully encourages the model to explore the discrete representation space far more effectively. By leveraging a richer vocabulary of learned features instead of relying on a small subset of popular codes, the model can capture more nuanced patterns within the data.

| Usage rate ↑ | Computer | CiteSeer | Cora | PubMed | Average |
|---|---|---|---|---|---|
| VQGraph | $17.9_{\blacktriangle 0.0}$ | $1.0_{\blacktriangle 0.0}$ | $1.9_{\blacktriangle 0.0}$ | $21.2_{\blacktriangle 0.0}$ | 10.5 |
| **DualVC**$_{GAT}$ | $30.5_{\blacktriangle 12.6}$ | $2.3_{\blacktriangle 1.3}$ | $4.6_{\blacktriangle 2.7}$ | $29.3_{\blacktriangle 8.1}$ | 16.7 |
| **DualVC**$_{GCN}$ | $33.9_{\blacktriangle 16.0}$ | $2.7_{\blacktriangle 1.7}$ | $5.1_{\blacktriangle 3.2}$ | $28.1_{\blacktriangle 6.9}$ | **17.5** |

**Table 4: Comparison of codebook usage rates (%).** We report the percentage of codebook entries used for each model. Markers indicate improvement over VQGraph.

**Alignment with Intrinsic Graph Topology.** This increased representational diversity is not arbitrary; it directly translates to a better understanding of the graph's intrinsic community structure. To verify this, we compute the **cut value**, a metric that measures the alignment between a model's class predictions and the graph's topology (where a higher value indicates that nodes with the same prediction are more densely connected). As shown in **Table 5**, DualVC consis-

| Method | Computer | Citeseer | Cora | PubMed | Average |
|---|---|---|---|---|---|
| SAGE | $0.8951_{\blacktriangle 0.00}$ | $0.9535_{\blacktriangle 0.00}$ | $0.9385_{\blacktriangle 0.00}$ | $0.9597_{\blacktriangle 0.00}$ | 0.9367 |
| MLP | $0.6764_{\blacktriangledown 0.22}$ | $0.8107_{\blacktriangledown 0.14}$ | $0.7203_{\blacktriangledown 0.22}$ | $0.9062_{\blacktriangledown 0.05}$ | 0.7784 |
| GLNN | $0.8579_{\blacktriangledown 0.04}$ | $0.9447_{\blacktriangledown 0.01}$ | $0.8908_{\blacktriangledown 0.05}$ | $0.9298_{\blacktriangledown 0.03}$ | 0.9058 |
| NOSMOG | $0.9047_{\blacktriangle 0.01}$ | $0.9659_{\blacktriangle 0.01}$ | $0.9480_{\blacktriangle 0.01}$ | $0.9641_{\blacktriangle 0.00}$ | 0.9457 |
| VQGraph | $0.9190_{\blacktriangle 0.02}$ | $0.9786_{\blacktriangle 0.03}$ | $0.9684_{\blacktriangle 0.03}$ | $0.9883_{\blacktriangle 0.03}$ | 0.9636 |
| **DualVC** | $0.9217_{\blacktriangle 0.03}$ | $0.9830_{\blacktriangle 0.03}$ | $0.9714_{\blacktriangle 0.03}$ | $0.9934_{\blacktriangle 0.03}$ | **0.9674** |

**Table 5: Comparison of the cut value.** The table shows that DualVC predictions are highly consistent with the graph topology, outperforming other methods. Markers indicate improvement or regression relative to SAGE. The best result in each column is in **bold**.

tently achieves the highest cut value across all evaluated datasets. This provides strong quantitative evidence that the diverse features learned by the dual codebooks are highly coherent with the underlying graph topology. This strong alignment is a key factor in its state-of-the-art classification performance, as it indicates the model is learning structurally meaningful representations.

## 6    ABLATION STUDIES AND MODEL ROBUSTNESS

Finally, we perform a series of ablation studies to validate the contribution of key architectural components and assess the model's general robustness.

**Importance of the VQ Commitment Loss.** The commitment loss is a critical component for stabilizing the joint training of the GNN encoder and the VQ codebooks. To quantify its impact, we trained a variant of DualVC with this loss term removed. The results, presented in **Table 6**, show a significant and consistent drop

| Method | Citeseer | PubMed | Cora | Computer |
|---|---|---|---|---|
| w/o $\mathcal{L}_{commit}$ | $75.93 \pm 2.01$ | $85.21 \pm 0.59$ | $86.34 \pm 1.91$ | $87.98 \pm 0.74$ |
| **DualVC** | $\textbf{78.67} \pm \textbf{1.49}$ | $\textbf{87.25} \pm \textbf{0.68}$ | $\textbf{89.34} \pm \textbf{1.80}$ | $\textbf{91.22} \pm \textbf{0.83}$ |

**Table 6:** Ablation study on the impact of the commitment loss ($\mathcal{L}_{commit}$). Removing it degrades performance across all datasets, highlighting its importance.

in performance. For instance, accuracy on Cora and Amazon Computers falls by **3.0** and **3.24** absolute percentage points, respectively. This confirms that the commitment loss is essential for providing a stable regularization signal that enables the encoder to produce high-quality embeddings that can be effectively quantized by the codebooks. Without it, the encoder and codebooks can drift apart, leading to a suboptimal learning process.

**Sensitivity to Key Hyperparameters.**    In **Figure 3**, we analyze DualVC's performance across a range of values for four critical hyperparameters on the Cora dataset. The results demonstrate that the model is highly robust. As seen in Figure 3(a), performance is remarkably stable for codebook sizes ranging from 512 to 2048, indicating that the model is not dependent on a massive discrete space to achieve high performance. Figure 3(b) shows that the model benefits from a mild contrastive temperature, peaking around 0.125. Most importantly, Figures 3(c) and 3(d) confirm that our stochastic noise injection is a key mechanism for promoting codebook diversity; small amounts of noise (e.g., a standard deviation of 0.05-0.10) boost performance over the no-noise baseline, while performance gracefully degrades with excessive noise. This overall stability underscores DualVC's reliability and practicality for real-world deployment.

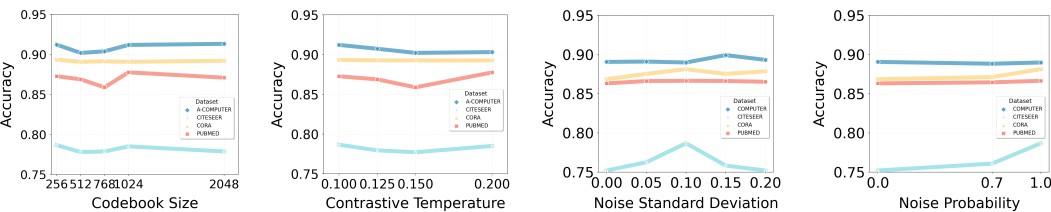

Figure 3: Comparison of (a) codebook size, (b) contrastive temperature, (c) noise standard deviation, and (d) noise probability vs. Accuracy.

## 7    CONCLUSION

In this work, we introduced DualVC, a novel Graph Contrastive Learning framework designed to address the limitations of noise and redundancy in continuous GNN embeddings. By generating two distinct views through parallel, stochastically perturbed vector-quantized codebooks, DualVC learns to extract essential and robust node features directly from a single source embedding. Our comprehensive experiments demonstrate that this approach sets a new state-of-the-art on multiple graph learning benchmarks, significantly outperforming prior methods with high computational efficiency. We showed that this superior performance is driven by the dual codebooks learning a more diverse set of features that are highly aligned with the underlying graph topology. Ultimately, DualVC validates the significant potential of using multiple, learned discrete bottlenecks for representation-level view generation, opening a promising new direction for self-supervised learning on graphs.

ETHICS STATEMENT

Our work focuses on developing more robust and efficient methods for self-supervised graph representation learning. The primary goal is to advance fundamental machine learning capabilities, which can reduce the computational resources required for training powerful graph models and improve performance on beneficial downstream applications, such as in bioinformatics, social network analysis, and recommendation systems. We acknowledge that the underlying training process requires significant computing resources, which contributes to energy consumption. Furthermore, we recognize that graph representation learning techniques, like any powerful AI technology, could be applied to domains with negative societal impacts, such as analyzing networks for malicious purposes. Our work does not use any personally identifiable information and relies on publicly available benchmark datasets. We advocate for the responsible application of graph learning and encourage ongoing research into the fairness, privacy, and transparency of these models to mitigate potential risks.

REPRODUCIBILITY STATEMENT

To ensure the reproducibility of our research, this paper provides a detailed account of our methodology and experimental setup. The core components of our DualVC framework, including the GNN encoder architecture, the dual vector-quantization layers, and the contrastive learning objective, are described in Section 4. Our complete experimental protocol, including dataset statistics, baseline implementations, evaluation procedures, and all hyperparameters, is presented in Section 5 and further detailed in Section C . The source code for our model, training scripts, and instructions to replicate our results are available at `https://anonymous.4open.science/r/DualVC-43C3/` to facilitate full verification of our findings.

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

# A    NOTATION SUMMARY

To aid in clarity and readability, we provide a summary of the key mathematical notations used in this paper. **Table 7** lists the symbols and their corresponding descriptions.

| Notation | Description |
|---|---|
| $\mathcal{G}$ | An input graph, defined as $\mathcal{G} = (\mathcal{V}, \mathcal{E})$. |
| $\mathcal{V}, \mathcal{E}$ | The set of nodes and edges in the graph, respectively. |
| $N$ | The total number of nodes in the graph, $|\mathcal{V}|$. |
| $X$ | The node feature matrix of size $N \times F_{in}$. |
| $A$ | The adjacency matrix of the graph. |
| $f_{enc}$ | The GNN encoder function with parameters $\theta_{enc}$. |
| $H$ | The matrix of continuous node embeddings from the GNN, $H \in \mathbb{R}^{N \times D_{gnn}}$. |
| $\mathbf{h}_i$ | The continuous embedding vector for node $i$. |
| $\tilde{\mathbf{h}}_{k,i}$ | The noise-perturbed embedding for node $i$ in channel $k$. |
| $\boldsymbol{\epsilon}_{k,i}$ | Gaussian noise vector added to the embedding for node $i$ in channel $k$. |
| $C_k$ | The learnable codebook for VQ channel $k$, where $k \in \{1, 2\}$. |
| $M$ | The number of vectors (codes) in each codebook. |
| $\mathbf{c}_{k,j}$ | The $j$-th codebook vector in codebook $C_k$. |
| $Q_k$ | The matrix of quantized node representations from VQ channel $k$. |
| $\mathbf{q}_{k,i}$ | The quantized representation vector for node $i$ from channel $k$. |
| $g_k$ | The non-linear projection head for VQ channel $k$. |
| $\mathbf{z}_{k,i}$ | The projected representation of node $i$ from channel $k$, used in the contrastive loss. |
| $\mathcal{L}_{\text{contrast}}$ | The main NT-Xent contrastive loss function. |
| $\mathcal{L}_{\text{VQ}}^{(k)}$ | The codebook loss for VQ channel $k$. |
| $\mathcal{L}_{\text{commit}}^{(k)}$ | The commitment loss for VQ channel $k$. |
| $\mathcal{L}_{\text{DualVC}}$ | The final, overall objective function for the DualVC model. |
| $\tau$ | The temperature hyperparameter in the contrastive loss. |
| $\beta$ | The weighting hyperparameter for the commitment loss. |
| $\lambda_{\text{commit}}$ | The weighting hyperparameter for the total commitment loss in the final objective. |
| $\text{sg}[\cdot]$ | The stop-gradient operator. |
| $\text{sim}(\cdot, \cdot)$ | The cosine similarity function. |

Table 7: Summary of notations used in the paper.

# B    DATASET DETAILS

We evaluate our model on six standard graph benchmarks that are widely used in the graph representation learning literature. A detailed description of each dataset is provided below, and their key statistics are summarized in **Table 8**.

- **Cora (McCallum et al., 2000), Citeseer (Giles et al., 1998), and PubMed (Canese & Weis):** These are citation network datasets where nodes represent academic publications and edges represent citation links between them. The node features are derived from the textual content of the papers, typically using a bag-of-words representation of the title and abstract. These datasets are common benchmarks for node classification and link prediction tasks due to their high-dimensional, sparse feature spaces and clear community structures.

- **Amazon Photo and Amazon Computers (Shchur et al., 2018):** These datasets are co-purchase graphs extracted from the Amazon e-commerce platform. Nodes represent products, and an edge exists between two nodes if the corresponding products are frequently bought together. The node features are constructed from bag-of-words representations of product reviews, and the class labels correspond to the product categories. These graphs are denser and larger than the citation networks, presenting a different set of challenges.

- **ogbn-arxiv (Wang et al., 2020):** The Open Graph Benchmark (OGB) arXiv dataset is a large-scale citation graph representing all Computer Science (CS) papers on the arXiv preprint server. Nodes are papers, and a directed edge from paper A to paper B indicates that

A cites B. The node features are 128-dimensional embeddings derived from the paper's title and abstract using a pre-trained language model. Its scale makes it a challenging benchmark for testing the efficiency and scalability of graph learning models.

| Dataset | Nodes | Edges | Classes | Features |
|---|---|---|---|---|
| Cora | 2,708 | 5,278 | 7 | 1,433 |
| Citeseer | 3,327 | 4,552 | 6 | 3,703 |
| Pubmed | 19,717 | 44,324 | 3 | 500 |
| Amazon-Computers | 13,752 | 491,722 | 10 | 767 |
| Amazon-Photo | 7,650 | 238,162 | 8 | 745 |
| ogbn-arxiv | 169,343 | 1,166,243 | 40 | 128 |

Table 8: Key statistics for the datasets used in our experiments.

## C    IMPLEMENTATION AND HYPERPARAMETER SETTINGS

All experiments were conducted using the PyTorch and DGL libraries on a server equipped with 8 NVIDIA A6000 GPUs. For our proposed DualVC model, we performed a thorough hyperparameter search to ensure optimal performance on each dataset. We separate the settings into two categories: dataset-specific tuned parameters and global parameters that were fixed across all experiments.

**Tuned Hyperparameters.**    The main hyperparameters, including learning rate (LR), codebook size, GNN dimensions, contrastive temperature, and commitment loss weight, were tuned for each dataset. The optimal configurations found through our search are detailed in **Table 9**. The codebook size is reported as '(size per codebook) x 2', reflecting our dual-codebook architecture. The "Hidden/Output" column refers to both the hidden dimension and the final output dimension of the GNN encoder, which were kept consistent.

| Dataset | LR | Codebook Size | Hidden/Output | VQ Dim | Temp. | Commit |
|---|---|---|---|---|---|---|
| Cora | 0.0030 | 256×2 | 512 | 512 | 0.10 | 0.1 |
| Citeseer | 0.0020 | 256×2 | 512 | 512 | 0.10 | 0.1 |
| Pubmed | 0.0005 | 1024×2 | 1024 | 1024 | 0.20 | 0.1 |
| Amazon-Computers | 0.0005 | 2048×2 | 512 | 512 | 0.20 | 0.1 |
| Amazon-Photo | 0.0005 | 256×2 | 512 | 512 | 0.125 | 0.1 |
| Arxiv | 0.0001 | 512×2 | 512 | 512 | 0.05 | 0.1 |

Table 9: Dataset-specific tuned hyperparameters for the DualVC model.

**Global Hyperparameters.**    Several hyperparameters related to the training procedure and model architecture were kept constant across all datasets to ensure a fair and consistent experimental setup. These default settings are listed in **Table 10**. We used the Adam optimizer for all training runs. The projection head is a 2-layer MLP with a ReLU activation function and a hidden dimension of 512 units. For the stochastic noise injection, which we identified as a key component for promoting codebook diversity, we used Gaussian noise with a standard deviation of 0.05, applied to all embeddings (probability of 1.0). Early stopping was employed with a patience of 20 epochs to prevent overfitting.

| Parameter | Value |
|---|---|
| Optimizer | Adam |
| Batch size | 256 |
| Epochs | 600 |
| Patience for early stopping | 20 |
| EMA decay for codebook updates | 0.99 |
| Codebook refresh threshold | 2 |
| Noise injection | Gaussian, $\sigma = 0.05$, prob.=1.0 |
| Projection head | 2-layer MLP (ReLU, 512 units) |

Table 10: Global default hyperparameters used across all datasets.

# D  FORMAL ANALYSIS OF THE DUALVC OPTIMIZATION PROCESS

This section provides a mathematical framework for the optimization of the DualVC model. We formalize the objective function and derive the gradients for all trainable parameters using a series of definitions, theorems, and proofs.

### D.0.1  DEFINITIONS AND PRELIMINARIES

We begin by formally defining the core components of the objective function.

**Definition D.1** (The DualVC Objective Function). Let $\mathcal{D} = (\mathcal{G}, X)$ be the input graph data. Let the set of all trainable parameters be $\Theta = \{\theta_{enc}, \theta_{g1}, \theta_{g2}, C_1, C_2\}$. The learning process aims to find $\Theta^*$ that minimizes the objective function $\mathcal{L}_{\text{DualVC}}$. The objective is composed of three distinct loss functions.

First, the **Contrastive Loss** ($\mathcal{L}_{\textbf{contrast}}$) is defined as:

$$\mathcal{L}_{\text{contrast}}(Z_1, Z_2) = -\frac{1}{2N} \sum_{i=1}^{N} \left[ \log \frac{\exp(\text{sim}(\mathbf{z}_{1,i}, \mathbf{z}_{2,i})/\tau)}{\sum_{j=1}^{N} \exp(\text{sim}(\mathbf{z}_{1,i}, \mathbf{z}_{2,j})/\tau)} + \log \frac{\exp(\text{sim}(\mathbf{z}_{2,i}, \mathbf{z}_{1,i})/\tau)}{\sum_{j=1}^{N} \exp(\text{sim}(\mathbf{z}_{2,i}, \mathbf{z}_{1,j})/\tau)} \right]$$

Second, the **Commitment Loss** ($\mathcal{L}_{\textbf{commit}}$) for each VQ channel $k \in \{1, 2\}$ is given by:

$$\mathcal{L}_{\text{commit}}^{(k)}(H, Q_k) = \beta \sum_{i=1}^{N} \sum_{d=1}^{D_{gnn}} (h_{i,d} - \text{sg}[q_{k,i,d}])^2 = \beta \|H - \text{sg}[Q_k]\|_F^2$$

Third, the **Codebook Loss** ($\mathcal{L}_{\textbf{VQ}}$) for each channel is:

$$\mathcal{L}_{\text{VQ}}^{(k)}(H, Q_k) = \sum_{i=1}^{N} \sum_{d=1}^{D_{vq}} (\text{sg}[h_{i,d}] - q_{k,i,d})^2 = \|\text{sg}[H] - Q_k\|_F^2$$

The main objective function for updating the encoder and projection heads is:

$$\mathcal{L}_{\text{DualVC}} = \mathcal{L}_{\text{contrast}} + \lambda_{\text{commit}}(\mathcal{L}_{\text{commit}}^{(1)} + \mathcal{L}_{\text{commit}}^{(2)})$$

**Definition D.2** (Vector Quantization and the Gradient Estimator). The vector quantization operator $Q_k(\cdot)$ maps a continuous input embedding $\mathbf{h}_i$ to the closest vector in codebook $C_k$:

$$\mathbf{q}_{k,i} = Q_k(\mathbf{h}_i) = \mathbf{c}_{k,j^*} \quad \text{where} \quad j^* = \arg\min_{j \in \{1,...,M\}} \|\mathbf{h}_i - \mathbf{c}_{k,j}\|_2^2$$

As the $\arg\min$ operator is non-differentiable, we introduce the Straight-Through Estimator (STE) as an axiom for our derivations. The gradient of the quantization operator with respect to its input is approximated as the identity matrix:

$$\nabla_{\mathbf{h}_i} Q_k(\mathbf{h}_i) \approx I$$

### D.0.2  THEOREMS AND PROOFS OF GRADIENT FLOW

With these definitions, we can now state and prove theorems regarding the gradient flow to each model component.

**Theorem D.1** (Gradient with Respect to Encoder Output). *The partial derivative of the total objective function $\mathcal{L}_{DualVC}$ with respect to the GNN encoder's continuous output matrix $H$ is given by:*

$$\frac{\partial \mathcal{L}_{DualVC}}{\partial H} = \sum_{k=1}^{2} \left( \frac{\partial Z_k}{\partial Q_k} \frac{\partial \mathcal{L}_{contrast}}{\partial Z_k} \right) + 2\lambda_{commit}\beta \sum_{k=1}^{2} (H - sg[Q_k])$$

*Proof.* The total derivative can be decomposed by linearity:

$$\frac{\partial \mathcal{L}_{\text{DualVC}}}{\partial H} = \frac{\partial \mathcal{L}_{\text{contrast}}}{\partial H} + \lambda_{\text{commit}} \sum_{k=1}^{2} \frac{\partial \mathcal{L}_{\text{commit}}^{(k)}}{\partial H}$$

We will prove the form of each term in two lemmas. $\qquad \square$

**Lemma D.2** (Gradient of the Commitment Loss).

$$\frac{\partial \mathcal{L}_{commit}^{(k)}}{\partial H} = 2\beta(H - sg[Q_k])$$

*Proof.* By Definition 1, $\mathcal{L}_{commit}^{(k)} = \beta\|H - sg[Q_k]\|_F^2$. The stop-gradient operator treats $Q_k$ as a constant with respect to $H$. The derivative of the squared Frobenius norm with respect to the matrix $H$ is:

$$\frac{\partial}{\partial H}\left(\beta\|H - sg[Q_k]\|_F^2\right) = \beta \cdot 2(H - sg[Q_k]) = 2\beta(H - sg[Q_k])$$

This completes the proof of the lemma. $\qquad\square$

**Lemma D.3** (Gradient of the Contrastive Loss).

$$\frac{\partial \mathcal{L}_{contrast}}{\partial H} \approx \sum_{k=1}^{2}\left(\frac{\partial Z_k}{\partial Q_k}\frac{\partial \mathcal{L}_{contrast}}{\partial Z_k}\right)$$

*Proof.* The contrastive loss $\mathcal{L}_{contrast}$ is not a direct function of $H$. We must apply the chain rule through the quantization layers ($Q_k$) and projection heads ($Z_k = g_k(Q_k)$).

$$\frac{\partial \mathcal{L}_{contrast}}{\partial H} = \sum_{k=1}^{2}\frac{\partial Q_k}{\partial H}\frac{\partial \mathcal{L}_{contrast}}{\partial Q_k}$$

Using the STE from Definition 2, we approximate $\frac{\partial Q_k}{\partial H} \approx I$. Therefore:

$$\frac{\partial \mathcal{L}_{contrast}}{\partial H} \approx \sum_{k=1}^{2}\frac{\partial \mathcal{L}_{contrast}}{\partial Q_k}$$

Applying the chain rule again for the projection head $g_k$:

$$\frac{\partial \mathcal{L}_{contrast}}{\partial Q_k} = \frac{\partial Z_k}{\partial Q_k}\frac{\partial \mathcal{L}_{contrast}}{\partial Z_k}$$

Substituting this back completes the proof of the lemma. $\qquad\square$

By combining the results of Lemma 1.1 and Lemma 1.2, the proof of Theorem 1 is complete.

**Corollary D.3.1** (Gradient for GNN Encoder Parameters $\theta_{enc}$). *The gradient of the objective with respect to the GNN parameters $\theta_{enc}$ is:*

$$\frac{\partial \mathcal{L}_{DualVC}}{\partial \theta_{enc}} = \frac{\partial H}{\partial \theta_{enc}}\frac{\partial \mathcal{L}_{DualVC}}{\partial H}$$

*Proof.* This follows directly from applying the multivariate chain rule, where $H = f_{enc}(X, A; \theta_{enc})$, and using the result of Theorem 1. $\qquad\square$

**Theorem D.4** (Gradient for Projection Head Parameters $\theta_{gk}$). *The gradient of the objective with respect to the parameters $\theta_{gk}$ of a projection head $g_k$ is:*

$$\frac{\partial \mathcal{L}_{DualVC}}{\partial \theta_{gk}} = \frac{\partial Z_k}{\partial \theta_{gk}}\frac{\partial \mathcal{L}_{contrast}}{\partial Z_k}$$

*Proof.* The projection head parameters $\theta_{gk}$ only influence the final objective through the contrastive loss term. The commitment loss terms do not depend on $\theta_{gk}$. Therefore:

$$\frac{\partial \mathcal{L}_{DualVC}}{\partial \theta_{gk}} = \frac{\partial \mathcal{L}_{contrast}}{\partial \theta_{gk}}$$

By the chain rule, since $Z_k = g_k(Q_k; \theta_{gk})$:

$$\frac{\partial \mathcal{L}_{contrast}}{\partial \theta_{gk}} = \frac{\partial Z_k}{\partial \theta_{gk}}\frac{\partial \mathcal{L}_{contrast}}{\partial Z_k}$$

The term $\frac{\partial Z_k}{\partial \theta_{gk}}$ is computed via standard backpropagation through the MLP $g_k$. $\qquad\square$

**Theorem D.5** (Optimal Codebook Update Condition). *The gradient of the codebook loss $\mathcal{L}_{VQ}^{(k)}$ with respect to a single codebook vector $\mathbf{c}_{k,j}$ is:*

$$\frac{\partial \mathcal{L}_{VQ}^{(k)}}{\partial \mathbf{c}_{k,j}} = 2 \sum_{i \in \mathcal{I}_j} (\mathbf{c}_{k,j} - sg[\mathbf{h}_i])$$

*where $\mathcal{I}_j = \{i \mid Q_k(\mathbf{h}_i) = \mathbf{c}_{k,j}\}$ is the set of indices of encoder outputs mapped to $\mathbf{c}_{k,j}$.*

*Proof.* From Definition 1, $\mathcal{L}_{VQ}^{(k)} = \sum_{i=1}^{N} \|sg[\mathbf{h}_i] - \mathbf{q}_{k,i}\|_2^2$. The gradient with respect to a specific vector $\mathbf{c}_{k,j}$ is non-zero only for terms where $\mathbf{q}_{k,i} = \mathbf{c}_{k,j}$.

$$\frac{\partial \mathcal{L}_{VQ}^{(k)}}{\partial \mathbf{c}_{k,j}} = \frac{\partial}{\partial \mathbf{c}_{k,j}} \sum_{i=1}^{N} \|sg[\mathbf{h}_i] - \mathbf{q}_{k,i}\|_2^2$$

$$= \sum_{i \in \mathcal{I}_j} \frac{\partial}{\partial \mathbf{c}_{k,j}} \|sg[\mathbf{h}_i] - \mathbf{c}_{k,j}\|_2^2$$

$$= \sum_{i \in \mathcal{I}_j} \frac{\partial}{\partial \mathbf{c}_{k,j}} \left( (sg[\mathbf{h}_i] - \mathbf{c}_{k,j})^T (sg[\mathbf{h}_i] - \mathbf{c}_{k,j}) \right)$$

$$= \sum_{i \in \mathcal{I}_j} 2(\mathbf{c}_{k,j} - sg[\mathbf{h}_i])$$

This completes the proof. □

**Corollary D.5.1** (Centroid Condition). *The gradient in Theorem 3 is zero if and only if the codebook vector $\mathbf{c}_{k,j}$ is the geometric centroid of its assigned encoder output vectors:*

$$\mathbf{c}_{k,j} = \frac{1}{|\mathcal{I}_j|} \sum_{i \in \mathcal{I}_j} sg[\mathbf{h}_i]$$

*Proof.* Setting the gradient from Theorem 3 to zero:

$$2 \sum_{i \in \mathcal{I}_j} (\mathbf{c}_{k,j} - sg[\mathbf{h}_i]) = 0 \implies |\mathcal{I}_j| \mathbf{c}_{k,j} = \sum_{i \in \mathcal{I}_j} sg[\mathbf{h}_i]$$

Dividing by $|\mathcal{I}_j|$ yields the result. This shows that the gradient-based update moves each codebook vector towards the mean of its assigned input vectors. □

## E  COMPUTATIONAL RESOURCES

All experiments, including model pre-training and downstream evaluation, were conducted on a server equipped with 8 NVIDIA A6000 GPUs, each with 48GB of VRAM. The specific frameworks used were PyTorch and the Deep Graph Library (DGL).

## F  THE USE OF LARGE LANGUAGE MODELS (LLMs)

The research and intellectual contributions presented in this work, including the DualVC framework, its theoretical underpinnings, the design of experiments, and the analysis of results, were completed entirely by the authors. For the preparation of this manuscript, we used large language models (LLMs) exclusively for proofreading and to help improve the clarity and grammar of the text. LLMs were not used for generating any of the core ideas, methodology, or experimental outcomes.

