# OpenReview forum: "Two Lenses are Better Than One: Dual Vector Quantization for Self-Supervised Graph Learning"
_ICLR.cc/2026/Conference — ICLR 2026 Conference Withdrawn Submission_

### Official Review · Reviewer_veKv · 2025-10-24

**Soundness:** 3
**Presentation:** 3
**Contribution:** 3
**Rating:** 8
**Confidence:** 5

**Summary:**

This paper introduces DualVC for self-supervised graph representation learning that reframes the view generation process in Graph Contrastive Learning (GCL). Instead of relying on traditional data augmentations like edge dropping or feature masking, which can degrade graph structure, the authors propose generating contrastive views at the representation level. The core mechanism of DualVC involves encoding an input graph with a single GNN encoder to produce continuous node embeddings, and passing these embeddings through two parallel, independent vector-quantized (VQ) layers, each with its own learnable codebook. The authors hypothesize that contrasting these two distinct "discretization perspectives" forces the model to learn features that are invariant to quantization noise, thereby filtering redundancy and capturing more robust and compact information.

**Strengths:**

1) The core idea of using dual parallel VQ codebooks to generate contrastive views from a single GNN embedding. It cleverly sidesteps the primary challenge of standard GCL as that the reliance on heuristic data augmentations that risk corrupting the graph’s structural integrity.
2) DualVC achieves new SOTA results across all six node classification benchmarks (Table 1) and all four link prediction benchmarks (Table 3), especially when compared to the most relevant single-codebook VQGraph.
3) The analysis (Table 4) demonstrates that DualVC utilizes a significantly higher percentage of its codebook entries compared to a single-codebook model, suggesting it successfully avoids codebook collapse and learns a richer discrete vocabulary.
4) The hyperparameter sweep (Fig. 3c) clearly shows that performance improves with a small amount of Gaussian noise and then degrades, strongly supporting the authors' hypothesis that stochasticity is essential for encouraging diverse and robust representations.

**Weaknesses:**

1) This paper frames the VQ layers as the core view generation mechanism. However, the stochastic Gaussian noise added before quantization appears to be just as critical. The ablation in Figure 3c shows that performance at zero noise is suboptimal. So, is the dual VQ structure the key, or is the stochastic perturbation the key?
2) This paper should discuss and compare against other augmentation-free GCL methods, such as those that apply perturbations directly in the embedding space.
3) The method introduces the codebook size ($M$) as a new hyperparameter that must be tuned per dataset (ranging from 256 to 2048 in Table 9). This adds tuning complexity compared to standard GCL models.
4) Sec.4 should explicitly state how the codebooks $C_k$ are updated? Minimizing the loss $\mathcal{L}_{VQ}$ separately, or using an EMA update as is common in VQ-VAE literature?

**Questions:**

1) To follow up on Weakness-1, what happens if you use no noise ($\sigma=0$) but rely on two different random initializations for codebooks $C_1$ and $C_2$? Do they still collapse to a redundant solution?
2) The original VQGraph paper focused on GNN-to-MLP distillation. Is the "VQGraph" baseline in your Table 1 a re-implementation of that paper’s method, or is it your own contrastive baseline that uses only a single VQ codebook?
3) This paper uses two separate projection heads, $g_1$ and $g_2$. Did you experiment with a shared projection head as $g_1 = g_2$? Would this lead to representational collapse, like siamese networks?
4) The DualVC model for Arxiv uses a 512-dim hidden space, which seems standard. Does the VQ bottleneck reduce the memory footprint during training in a way that other models do not? Because, in Table 1, several strong GCL baselines (DGI, GRACE, GCA) are marked as OOM (Out of Memory) on the ogbn-arxiv dataset, while DualVC not.

---

### Official Review · Reviewer_NQgD · 2025-10-29

**Soundness:** 2
**Presentation:** 2
**Contribution:** 2
**Rating:** 2
**Confidence:** 5

**Summary:**

This paper presents DualVC,  which is a graph contrastive learning framework that addresses limitations of continuous embeddings in self-supervised graph representation learning. Instead of using traditional data augmentation for view generation, DualVC employs two parallel, independent vector-quantized (VQ) codebooks to create discrete views from a single GNN embedding. Then the contrastive objective is employed on the learned codebooks of two views. The authors demonstrate state-of-the-art performance across six benchmark datasets.

**Strengths:**

1. The overall model design is simple and easy to follow.
2. Achieves state-of-the-art across 6 diverse benchmarks spanning different graph types and scales.

**Weaknesses:**

1. The motivation of this paper is weak. The introduction claims that continuous embeddings are “redundant/noisy” and hard to interpret, yet provides no quantitative evidence or citations demonstrating that these issues are practical bottlenecks in GCL. While random perturbation–based augmentations may indeed distort structural information, many studies have already proposed informative and structure-preserving augmentations, such as AD-GCL (Suresh et al., NeurIPS 2021) and SPAN (Lin et al., ICLR 2023). Moreover, the paper overlooks augmentation-free GCL approaches like GraphACL (Xiao et al., NeurIPS 2023), which directly address the claimed limitations. Finally, the experimental comparisons rely on overly simplistic baselines and fail to include several recent, stronger GCL methods, making the claimed improvements less convincing.
2. The paper doesn’t articulate what invariances each codebook is supposed to capture, nor why two is the right number (vs. one/hierarchical/multi-codebook mixtures).
3. If the codebook mapping defined in Eq. (4)–(6) were removed and the contrastive learning objective were directly applied to the noisy embeddings, the resulting model would closely resemble SimGRACE (Xia et al., WWW 2022). Therefore, a direct comparison with SimGRACE is necessary to demonstrate the effectiveness of the proposed VQ mechanism.
4. In Eq. (3), the authors add Gaussian noise to the node embeddings to generate distinct views. However, this design choice is unclear, because given that the two codebooks are initialized differently, they should already yield sufficiently distinct representations.
5. Downstream representations are the continuous encoder embeddings $H$ rather than discrete codes, which undercuts motivation around discrete/compact/structured latents improving interpretability or deployment. If discrete latents are the key, show they work at inference too.
6. Why two views are complementary? This property should ideally emerge from the model design itself, enforced by explicit constraints or objectives that encourage complementary codebooks. Moreover, theoretical analysis is needed to demonstrate and justify such complementarity.  Although in the experiments, usage rate is used as evidence, but it is a coarse occupancy statistic and does not prove complementarity.

**Questions:**

See weaknesses.

---

### Official Review · Reviewer_rqdv · 2025-11-01

**Soundness:** 3
**Presentation:** 3
**Contribution:** 3
**Rating:** 4
**Confidence:** 3

**Summary:**

The paper proposes DualVC: on the continuous representations produced by the same GNN encoder, two VQ (vector-quantization) codebooks are attached in parallel to obtain two discrete views; a contrastive loss is then used to align these two discrete views, training the entire model. For downstream tasks, the method does not use the discrete codes; instead, it directly uses the continuous GNN representation H as the final embedding for classification/link prediction.

**Strengths:**

1. Treating “discretization” directly as a “view generator,” create positives with two codebooks, parallel to common augmentation strategies; the approach is easy to implement and can pair with any GNN encoder. The method definition and overall objective are clear.
2. Results are provided on multiple benchmarks, achieving SOTA in many cases.

**Weaknesses:**

1. The method’s motivation emphasizes that discretization improves compactness and interpretability, yet the downstream evaluations uniformly use the continuous representation H (rather than the quantized Q1/Q2). Therefore, there is no direct validation of the usefulness/interpretability of the discrete representation itself, and the conclusions are insufficiently supported. The paper uses “codebook usage rate” and “cut value” as indirect evidence, but it does not directly show whether the semantics learned by the two codebooks are complementary (e.g., codeword–substructure correspondences, cross-codebook mutual information or decorrelation metrics, qualitative visualizations). At present, “higher usage rate” does not imply “more complementarity.”
2. The contrastive learning objective may degenerate into self-consistency regularization. The positive pair comes from two quantizations of the same encoder output (with high-probability Gaussian noise). Without graph augmentations or semantic perturbations, InfoNCE behaves more like a consistency constraint under quantization noise. The paper does not analyze the composition of negative samples or how batch size affects representation capacity and the avoidance of “encoding node IDs”.

3. The authors tune DualVC per dataset plus global defaults, but they do not describe the hyperparameter budget or search space for each baseline, creating a risk that the proposed method is thoroughly tuned while baselines rely on defaults.
4. On Amazon graphs, the near-perfect AUC/AP (AUC≈98.99/AP≈99.10) is unusually high. If pretraining used the full graph, test edges may have been seen during pretraining. The paper does not clarify whether pretraining was performed after removing test edges, which requires clarification and additional controls.

**Questions:**

Adding linear-probing results that directly use Q1/Q2 or their combination; or provide qualitative/quantitative evidence aligning codewords with graph substructures (mutual information, AM/CKA, codeword-cluster visualizations, etc.).

---

### Official Review · Reviewer_ZCwx · 2025-11-07

**Soundness:** 2
**Presentation:** 2
**Contribution:** 3
**Rating:** 2
**Confidence:** 4

**Summary:**

This paper introduces DualVC, a framework that learns two complementary quantized views of GNN node embeddings via dual vector-quantization (VQ) modules, aligned through an NT-Xent contrastive loss. The overall objective combines this contrastive term with an additional commitment loss that encourages the learned representations to couple with their respective “codebooks.” Conceptually, the method sits between standard graph contrastive learning and vector quantization approaches. The authors present a broad set of experiments demonstrating that DualVC outperforms methods from both families on node classification and link prediction tasks, often by a substantial margin.

The authors argue that the learned dual quantized representations—serving as the basis for contrastive learning—address limitations of both continuous GNN embeddings (e.g., high dimensionality, redundancy, and noise sensitivity) and existing vector quantization methods (e.g., limited representational diversity, lack of complementary structural encoding). A method that successfully combines the strengths of both paradigms would indeed represent a strong contribution. However, the paper lacks sufficient detail in explaining, formalizing, and validating how these dual quantized representations are actually produced—despite being central to the claimed contributions of DualVC.

In particular, the supposed “duality” arises, according to the text, from a stochastic perturbation applied to the learned GNN embeddings, which should enable the two VQ layers to learn distinct quantized representations of each graph. However, the implementation in the released code and the sources of variation actually driving these dual codebooks differ substantially from what is described. Despite strong empirical results, the mechanism that induces distinct quantizations is under-explained, which limits both interpretability and reproducibility.

**Strengths:**

- Novel and appealing idea: contrasting two quantized views learned on top of GNN embeddings, solving issues with GCL and Vector Quantization
- Very strong empirical results.


I think this paper should be rejected in its current form, as it requires substantial updates to clearly explain and justify the underlying mechanisms that enable learning dual vector quantizations. The contributions are novel and promising, but the manuscript does not adequately describe or align with the methods implemented in the released codebase. I would be open to revising my score if the authors can reconcile these discrepancies and provide clear justifications demonstrating that the necessary clarifications and revisions would not constitute substantial methodological changes.

**Weaknesses:**

- Mechanism gap: The paper does not clearly explain how the two VQ modules learn meaningfully distinct representations. In the code, the main sources of divergence seem to stem from internal VQ randomness (k-means initialization, optional Gumbel sampling) and separate projection heads, rather than from explicit or designed perturbations to the embeddings.
- Paper–code mismatch: Key implementation details such as k-means initialization and EMA-based codebook updates are central to the code but not mentioned in the paper. Conversely, Gaussian perturbations claimed in the text are not implemented. Gumbel sampling, if used, introduces randomness only in the code selection step, not as additive noise to embeddings.
- Missing ablations: There is no analysis of the necessity of k-means initialization or EMA updates, the role of independent vs. shared projection heads, sensitivity to Gumbel temperature, codebook size, EMA decay, or the weighting of commitment and codebook terms. The effects of deterministic vs. stochastic assignments are also not explored.

**Questions:**

- Gaussian perturbation vs. Gumbel noise: The paper states that Gaussian noise is added to the embeddings before quantization to create dual views, but this is not present in the code. Was Gaussian noise actually used in any experiments? If so, where was it applied, with what $\sigma$, and was this the same setup used for the noise sensitivity experiments?
- K-means initialization: Both VQ modules appear to initialize their codebooks using k-means on the same embeddings but with independent centroid samples. Why is this not discussed in the manuscript?
- Sources of duality: Given that vq1 and vq2 are called with identical inputs, what actually ensures their representations differ? Is divergence primarily due to (a) independent k-means initialization, (b) stochastic assignment (if temperature > 0), or (c) independent projection heads? Ablations isolating each factor would be very helpful.
- Cosine vs. Euclidean codebooks: `vq.py` appears to support both Euclidean (distance-based) and cosine (similarity-based) codebook lookups, yet all provided SOTA scripts enable `--vq_use_cosine_sim`. What motivated preferring cosine—e.g., scale invariance, gradient stability, or empirical gains? Were ablations run comparing the two metrics across datasets (accuracy, convergence speed, codebook utilization/collapse rates)? If Euclidean was tested, how were feature norms handled or normalized so that distances balance appropriately with the commitment and contrastive losses? Why is none of this detailed in the appendix?

---

### Note · Authors · 2025-11-21

I have read and agree with the venue's withdrawal policy on behalf of myself and my co-authors.